# Gene and protein expression of mTOR and LC3 in hepatocellular carcinoma, colorectal liver metastasis and "normal" liver tissues

**Marina Bortolami**[1]*, **Alessandra Comparato**[1ʘ], **Clara Benna**[2ʘ], **Andrea Errico**[1ʘ],
**Isacco Maretto**[2‡], **Salvatore Pucciarelli**[2‡], **Umberto Cillo**[2‡], **Fabio Farinati**[1ʘ]

**1** Department of Surgery, Oncology and Gastroenterology, DISCOG, School of Medicine, Gastroenterology Unit, University of Padova, Padova, Italy, **2** Department of Surgery, Oncology and Gastroenterology, DISCOG, School of Medicine, Surgical Unit, University of Padova, Padova, Italy

ʘ These authors contributed equally to this work.
‡ These authors also contributed equally to this work.
* marina.bortolami@unipd.it

## Abstract

The physiological role of autophagy in the progression of liver diseases is still debated. To understand the clinical relevance of autophagy in primary e secondary hepatic tumors, we analyzed the expression of mTOR (mammalian target of rapamycin), a key regulator of autophagy; Raptor (regulatory-associated protein of mTOR); ULK1 (Unc-51 like kinase 1) determinant in the autophagy initiation; LC3 (microtubule-associated protein 1A/1B-light chain 3), a specific marker of autophagosomes; and p62, a selective autophagy receptor. Samples from subjects with chronic hepatitis (n.58), cirrhosis (n.12), hepatocellular carcinoma (HCC, n.56), metastases (n.48) from colorectal cancer and hyperplasia or gallbladder stones (n.7), the latter considered as controls, were examined. Gene expression analysis was carried out in n.213 tissues by absolute q-PCR, while protein expression by Western Blot in n.191 lysates, including tumoral, surrounding tumoral and normal tissues. Nonparametric statistical tests were used for comparing expression levels in the above-mentioned groups. Subgroup analysis was performed considering viral infection and chemotherapy treatment. The mTOR transcriptional level was significantly lower in metastases compared to HCC (P = 0.0001). p-mTOR(Ser2448) and LC3II/LC3I protein levels were significantly higher in metastases compared to HCC (P = 0.008 and P<0.0001, respectively). ULK (Ser757) levels were significantly higher in HCC compared to metastases (P = 0.0002) while the HCV- and HBV- related HCC showed the highest p62 levels. Chemotherapy induced a down-regulation of the p-mTOR(Ser2448) in metastases and in non-tumor surrounding tissues in treated patients compared to untreated (P = 0.001 and P = 0.005, respectively). **Conclusions:** the different expression of proteins considered, owning their interaction and diverse tissue microenvironment, indicate an impairment of the autophagy flux in primary liver tumors that is critical for the promotion of tumorigenesis process and a coexistence of autophagy inhibition and activation mechanisms in secondary liver tumors. Differences in mTOR and LC3 transcripts emerged in tumor-free tissues, therefore particular attention should be considered in selecting the control group.

**Data Availability Statement:** All relevant data are within the paper and its Supporting Information files.

**Funding:** MB: 2017-prot. BIRD175781 University of Padova The funders had no role in study design, data collection and analysis, decision to publish, or preparation of the manuscript.

# Introduction

Autophagy is the cellular self-digestion pathway responsible for the degradation of damaged cell components in the lysosome [1,2], and its dysfunction has been implicated in many human diseases [3], including cancer [4,5]. The role of autophagy in cancer is complicated and context-dependent [6]. In the liver, most studies support the tumor-suppressive function of autophagy, where it acts primarily as a mechanism that protects the liver from tumor onset [7–9]. Research using mouse models shows that deletions of specific Atg genes (autophagy-related genes) induce a greater onset of hepatocellular carcinoma (HCC) [10]. Globally, HCC is the fifth most common cancer, and the second highest in terms of mortality, due to its poor prognosis [11]. The primary risk factor is cirrhosis of the liver; indeed, HCC occurs in cirrhotic liver in 80% of cases [12]. The most important etiologic agents are the hepatitis C virus (HCV) and chronic infection from hepatitis B virus (HBV) [13]. However, chronic alcohol abuse, exposure to dietary aflatoxins, and fatty liver disease (NAFLD) are also reported to precede the development of HCC [14]. HCC is a highly-resistant-type of cancer, and surgical resection, in addition to transplant, is the first-choice treatment for patients not affected by hepatic cirrhosis or extrahepatic metastases. The liver is also the main site of metastasis from colorectal cancer (CRC) [15], the third most common cancer in men, and the second most common in women worldwide [16]. Surgical resection is the best curative treatment for patients with liver metastases from CRC, which can be aided by adjuvant chemotherapy [17].

The physiological role of autophagy in liver diseases and liver cancer is still debated. Activation of the autophagic mechanism is due to mTOR (mammalian target of rapamycin) a central regulator of cell growth that plays a key role at the interface of the pathways that coordinately regulate the balance between cell growth and autophagy. mTOR is a serine/threonine kinase found in two protein complexes: mTORC1 and mTORC2, which have different substrates and functions [18,19]. The multiprotein complex of TORC1 consists of mTOR, mLST8 (mammalian lethal with SEC13 protein 8), Raptor (Rapamycin-sensitive adapter protein of mTOR), PRAS40 (proline rich Akt substrate of 40kDa) and Deptor (DEP domain TOR-binding protein).

mTOR is auto-phosphorylated at Ser2481 and the phosphorylation of Ser2448, located in the catalytic domain, by PI3 kinase/Akt signaling pathway, stimulates its protein synthesis activity [20–22]. Raptor is important for the kinase activity of mTORC1. When active, mTORC1 represses autophagy by its association with ULK1 (UNC-51 like kinase)-Atg13-FIP200 complex through the binding of Raptor with ULK1. Inactive mTORC1 dissociates from ULK complex, promoting ULK activity [23].

Once autophagy has been activated, a series of autophagy-related protein complexes modulate the formation of double membrane vesicles called autophagosomes, which capture cytoplasmic cargo [24]. LC3 (microtubule-associated protein 1A/1B-light chain 3) is a soluble protein distributed ubiquitously throughout mammalian tissues, and proteolytically cleaved by Atg 4 (autophagy-related genes) to form LC3-I. This form is conjugated to the lipid phosphatidylethanolamine, generating LC3-II, recruited for phagophore formation. The phagophore engulfs long-lived proteins and organelles to become an autophagosome. The closed autophagosome fuses with lysosome rich in hydrolases which degrade and recycle the cargo [25].

LC3-II is the only protein marker reliably associated with the successful formation of the completed autophagosome [26]. A second marker of autophagy is represented by p62, a multidomain protein interacting LC3II in the autophagosome where it is sequestered and degraded. Therefore, the p62 and LC3II turnover represents an index of autophagy flux (autophagic degradation activity) [27].

Experimental models employing animal livers or hepatic cell lines cultures have contributed significantly to our current knowledge of autophagy regulation and function, although data in

human liver samples are hampered by discrepancies between studies related to the selection of patients and methods used for the autophagy analysis. Therefore, for a better understanding of the clinical relevance of autophagy in the different stages of liver diseases, we analyzed the gene and protein expression of mTOR and LC3 and the protein expression of Raptor, ULK1 and p62 in a large number of samples obtained from patients with chronic disease, HCC, liver metastases from CRC and "histologically normal" liver tissues. Thus, a different modulation of autophagy in primary and secondary hepatic tumors and in tumor-free biopsies, such as tissues with chronic hepatitis and cirrhosis and "normal" liver will be identified.

## Materials and methods

### Patients and characteristics

The study involved n. 181 patients with HCV- and HBV-related chronic hepatitis (CH: n.58) and cirrhosis (CIRR: n.12), hepatocellular carcinoma (HCC: n.56), liver metastases (M: n.48) from colorectal cancer (CRC) and normal liver (CONT: n.7) from cholecystectomy and hyperplasia. All patients were recruited from the Department of Surgery, Oncology and Gastroenterology, DISCOG, University of Padua, School of Medicine, Italy. We extracted the clinico-pathological data of patients treated at our institution (University Hospital of Padua, North East of Italy) between January 2002 and December 2016, using a prospectively maintained database linked to our institutional biobank (Biobank of the First Surgical Clinic I-DISCOG). To be included in the study, each case had to meet the following requirements: 1) histologically confirmed diagnosis of HCC/CRC liver metastasis/chronic hepatitis/cirrhosis; 2) pathology-based information on TNM stage; 3) follow-up data (minimum follow up: six months); 4) availability of tumor tissues, tumor surrounding tissues and non-tumor tissues. Written informed consent was obtained from all patients included in the study. The University-Hospital Ethics Committee of Padua approved the study protocol (no.47081, CESC code 3312/AO/14).

The diagnosis of chronic HCV infection was obtained based on HCV-RNA positivity by polymerase chain reaction (PCR), persistent abnormality of transaminases for at least 12 months, and compatible histology. HBV-DNA was tested by a commercially available fluid phase hybridization assay (Abbott, Chicago-Illinois, USA). In HCV-infected patients, HCV–specific serum antibodies were detected by enzyme immunoassay tests (EIA-II; Ortho Diagnostic System) confirmed by recombinant immunoblot assay (RIBA-II; Ortho Diagnostic System) according to the manufacturer's instructions. HBV serum markers were tested by radio immune assay (RIA) (Abbott, Chicago-Illinois, USA). A single pathologist made the histologic diagnoses, and the score, according to Ishak's classification [28], was determined in all biopsy samples.

### Liver biopsies and tissue specimens

Liver biopsies were obtained during US-guided liver biopsy (Menghini modified needle 16–18 gauge) from patients with chronic hepatitis or liver cirrhosis. Biopsies were performed in accordance with a standard protocol. The tissue samples, obtained from HCC and liver metastases at the time of surgical resection, were macroscopically selected from the middle of the nodule. In patients with HCC, surgical specimens were obtained from the tumoral mass and non-tumoral tissue at least 1 cm from the tumor border of the resected specimen (when possible). From patients with liver metastases, the tissue samples were obtained from a metastatic nodule, tissue adjacent to the metastatic nodule, and far from the adjacent metastatic nodule tissue in 30/49 patients. All surgical liver specimens were dissected and snap frozen in liquid nitrogen and stored in liquid nitrogen at the Biobank of the First Surgical Clinic I-DISCOG. The routine histological analysis was performed in blind fashion by a pathologist.

## RNA extraction

Total RNA was extracted from frozen hepatic tissue by acid guanidium thiocyanate-phenol-chloroform. RNA concentration was spectrophotometrically quantified. The integrity of the RNA sample was assessed by electrophoresis on a 2% agarose gel (FMC Bio Product, Rockland, ME, USA). The quality of the isolate RNA was also assessed using RNA 600 Nano Assay and the Agilent 2100 bioanalyzer (Agilent technologies Palo Alto, CA, USA).

## Reverse transcription

For the synthesis of complementary DNA (cDNA), 1μg of RNA was reverse transcribed in the presence of 1X PCR Buffer, 1mM each of dNTPs (dATP, dTTP, dCTP, dGTP), 1U/μl RNase inhibitor, 2.5μM random hexamers, and 2.5U/μl of Murine Leukemia Virus (Perkin Elmer, Foster City, CA, USA). The reverse transcription reaction was performed at 25°C for 10 min, 42°C for 15 min, and 99°C for 5 min, and carried out in Perkin Elmer GeneAmp PCR System 2400.

## Primers

Oligonucleotide primers were designed with the Primer Express software (ABI/PE Applied Biosystems, Foster City, CA, USA) and synthetized by Eurofins Genomics (Ebersberg, D Milano, Italy). Nucleotide sequences for the sense and antisense primers used for real-time PCR were:

5'-GCCTTCTTCCTGCTGGTGAA-3', 5'-TCCTGCTCGTAGATGTCCGC-3'[ENST00000360668] for LC3 and the length of this amplicon was 74 bp; 5'-CTGCTTCCTCGGACAACC-3', 5'- GACAA CAGCCTTCTGGTGGC-3'[ENST00000569417] for mTOR and the length of this amplicon was 91 bp; 5'- CCTGGCACCCAGCACAA-3', 5'-CGATCCACACGGAGTACT-3' [ENST00000158302] for β-actin, and the length of this amplicon was 70 bp.

## PCR products analysis

The PCR products underwent vertical electrophoresis on a 0.75 mm thick, non-denaturing 6% acrylamide/bis-acrylamide gel with 5% glycerol. The silver nitrate stained band was analyzed using UVITEC Alliance 2M (Cambridge, UK).

## DNA purification

Purified DNAs were obtained using the MiniElute PCR Purification Kit according to the manufacturer's protocol. The concentration of the purified amplicons was quantified spectrophotometrically with the biophotometer 6131 (Eppendorf, Hamburg, Germany), and the PicoGreen dsDNA quantitation assay. Fluorescence was measured using 480 nm excitation and 520 nm emission (Luminescence Spectrometer, Perkin Elmer, Foster City, CA).

## Quantitative absolute real-time PCR

Real-time PCR was conducted in ABI 7900 Sequence Detection System (Applied Biosystems, Foster City, CA, USA) using SYBR Green I. PCR was done in a 25μL final volume containing 1x TaqMann buffer, 5.5mmol of $MgCl_2$, 200μmol of nucleotides, 0.01U/ml of AmpErase UNG, 0.25U of AmpliTaq Gold Polymerase (SYBR Green Master Mix), 300nM of each primer, and 5μL of cDNA template. After one 2-min step at 50°C and a second step at 95°C for 10 min, the samples underwent 45 cycles of 45 seconds at 94°C, followed by 45 seconds at 65°C for mTOR, 45 seconds at 67°C for LC3, and 45 seconds at 62°C for β-actin. A final extension step was performed at 60°C for 10 min. All determinations were performed in triplicate.

## Gene expression quantification

The amounts of mRNA in the unknown samples were determined from the standard curves of the gene of interest. Standard curves were generated using serial dilution (1:10) from $10^8$ to $10^2$ copies/μl of reference samples. The data was expressed as a ratio of the transcript amounts of the gene of interest to the β-actin transcript, the housekeeping gene.

## Western blotting

Tissue samples were homogenized in RIPA lysis buffer (20mM Tris-HCl pH 7.4; 150mM NaCl; 5 mM EDTA; 1.5% Niaproof; 1mM Sodium orthovanadate; 0.1% SDS; 0.5mM PMSF) with the addition of protease inhibitors (Chymotrypsin 1.5μg/ml, Termolysin 0.8μg/ml, Papain 1mg/ml, Pronase 1.5μg/ml, Pancreatic extract 15μg/ml, Trypsin 0.2μg/ml) (cOmplete™ Protease Inhibitor Cocktail, Roche). The homogenates were incubated on ice for 20 min, and then centrifuged at 13,000 rpm for 20 min at 4˚C. The supernatant was collected and stored at -80˚C. The "Pierce BCA Protein Assay Kit" (Bio-Rad, Hercules, CA, USA) was used for protein determination. Western blotting was performed by subjecting 20 μg total protein extracted to SDS–PAGE. For Western blot analyses, protein lysates were loaded onto 7.5% and 15% gradient gels, and run at 80 V. After electrophoresis, the proteins were transferred to a nitrocellulose membrane (0.22 mm Hybond ECL, GE Healthcare, Buckinghamshire, UK). The membranes were blocked with 5% fat-free milk in Tris-buffered saline containing 0.1% Tween-20. Membranes were probed with anti-human monoclonal antibodies (Cell Signaling, Danvers, MA, USA): phospho-mTOR (Ser2448) rabbit Ab [(D9C2) XP-R #5536] 1:1000; mTOR rabbit Ab (#2972) 1:1000; LC3A/B rabbit Ab [(D3U4) XP-R, #1274] 1:1000; phosphor-Raptor (Ser 792) Ab (#2083) 1:1000; Raptor rabbit Ab (#2280) 1:10000; phosphor-ULK1 (Ser757) Ab(#6888) 1:1000; ULK1 rabbit Ab [(D8H5) #80544] 1:1000; SQSTM1/p62 Ab (#5114)1:1000 and β-actin (D6A8, #8457) 1:1000. After incubation with the secondary anti-rabbit HRP-linked IgG antibody 1:3000, (Cell Signaling, Danvers, MA, USA), immunoreactive proteins were visualized with enhanced chemiluminescence reagents "ECL Fast Femto" (SuperSignal™). Molecular sizing was carried out using the Full-Range Rainbow Molecular Weight Marker (GE Healthcare, Buckinghamshire, UK).

Each band was quantified through densitometry using "UVITEC ALLIANCE MINI 2M" (Cambridge, UK), and the results presented as the ratio of the intensity of the band of phosphorylated and non-phosphorylated proteins, LC3-II and LC3-I, p62 and β-actin.

## Statistical analysis

We calculated the sample size assuming a difference between means of 25, considered relevant in our study, a standard deviation of 20 and a ratio between controls and cases of 1:4. The estimated sample size was 32 subjects, 7 for the healthy controls and 28 for the patients of each category. To reach the calculated samples size (in terms of RNA samples and protein samples), when possible, we doubled the enrolled patients. The power of the present analysis was calculated for each comparison between two means.

All results are given as mean values ± SD (standard deviation). Statistical analyses were performed using the R software (Free Software Foundation Cambridge, MA, USA) and GraphPad Prism 8.3.1 (GraphPad Software, San Diego, CA). The differences between the groups were analyzed with the Kruskal-Wallis test. The relationship between two variables was assessed by Spearman non-parametric correlation. The P-values of the comparisons were corrected for multiple testing (q-values) employing the False Discovery Rate (Benjamini and Hochberg 1995). q-values < 0.05 were considered significant. All statistical analyses were based on a two-sided hypothesis test.

## Results

### Gene expression

Etiological characteristics of the 181 enrolled subjects are listed in Table 1.

mTOR and LC3 gene expression analysis was done using n.213 RNAs extracted from: n.58 tissues of chronic hepatitis (CH); n.12 cirrhotic (CIRR); n.26 tissues surrounding HCC (PHCC); n.26 HCC; n.36 histologically normal liver tissue resected as far as possible from the metastatic nodule (NM); n.27 tissues surrounding the metastatic nodules (PM); n.23 metastatic nodule (M); n.5 controls (CTRL).

The power of the analyses is reported in S1 Table.

**mTOR Gene expression.** A statistically significant difference (P<0.0001, by Kruskal-Wallis test) in mTOR transcript levels among the groups was found.

No difference emerged in the mTOR transcript levels between CTRL (62.07±16) tissues and tissues obtained from patients with CH (53.3±30), CIRR (56.2±30) and HCC (51.6±37), but CTRL showed significantly higher mTOR transcript levels than PM (30.6±18; q = 0.04) and M (25.8±18; q = 0.008).

M tissues (25.8±18) showed the lowest mTOR transcript levels, which were statistically lower than in CH (q = 0.003), CIRR (q = 0.004), PHCC (39.7±18; q = 0.01), HCC (q = 0.005) and NM (37.4±15; q = 0.03) (Fig 1A).

When all tissues were considered according to the presence (Virus) or absence (Virus-) of viral infection, mTOR gene expression was significantly higher in both Virus HCC (58.3±44) and Virus- HCC (47.0±33) than in M (q = 0.01 and q = 0.04 respectively) (Fig 1C).

HCV-related CH tissues expressed the highest mTOR levels (61.7±35) which are statistically significant compared with PM (q = 0.009) and M (q = 0.003). A statistically significant difference was also found in the HCV-CIRR tissues (53.1±26) compared with M (q = 0.01) (Fig 1E).

mTOR mRNA levels in HBV-related CH (44.4±20) and HBV-related HCC (81.3±48) tissues were significantly higher than in M (q = 0.003 and q = 0.009 respectively) (Fig 1G).

According to pre-operative (CT) or no pre-operative (NA) chemotherapy no statistically significant difference was found between NM NA and CTRL tissues while the latter showed significantly higher levels compared with all other metastasis-related tissues (Fig 2A).

**Table 1. Etiological characteristics of the 181 enrolled subjects.**

| | | Controls | Chronic Hepatitis | Cirrhosis | HCC | Meta |
|---|---|---|---|---|---|---|
| n. Patients | | 7 | 58 | 12 | 56 | 48 |
| gender (%) | F | 6 (85.7) | 14 (24.1) | 3 (25.0) | 20 (35.7) | 19 (39.6) |
| | M | 1 (14.3) | 44 (75.9) | 9 (75.0) | 36 (64.3) | 29 (60.4) |
| Age, mean (st. dev) | | 50.00 (18.93) | 45.31 (12.98) | 50.75 (11.10) | 63.04 (11.54) | 61.55 (10.00) |
| Cause (%) | VIRUS | | 58 (100) | 12 (100) | 28 (60.9) | |
| | Alcohol | | | | 5 (10.9) | |
| | Crypto | | | | 8 (17.4) | |
| | no Cirr | | | | 4 (8.7) | |
| | META | | | | | 48 (100) |
| | HFE | | | | 1 (2.2) | |
| Virus (%) | no Virus | 7 (100) | | | 19 (33.3) | 48 (100) |
| | HBV | | 29 (50.0) | 2 (16.7) | 9 (15.8) | |
| | HCV | | 29 (50.0) | 10 (83.3) | 28 (49.1) | |
| Treatment (%) | NA | | | | | 10 (33.3) |
| | CT | | | | | 20 (66.7) |

Crypto: Cryptogenic; no Cirr: Without cirrhosis; META: Metastasis; HFE: Hemochromatosis; NA: Non adjuvant therapy before surgery for colorectal cancer; CT: Chemotherapy before surgery for colorectal cancer.

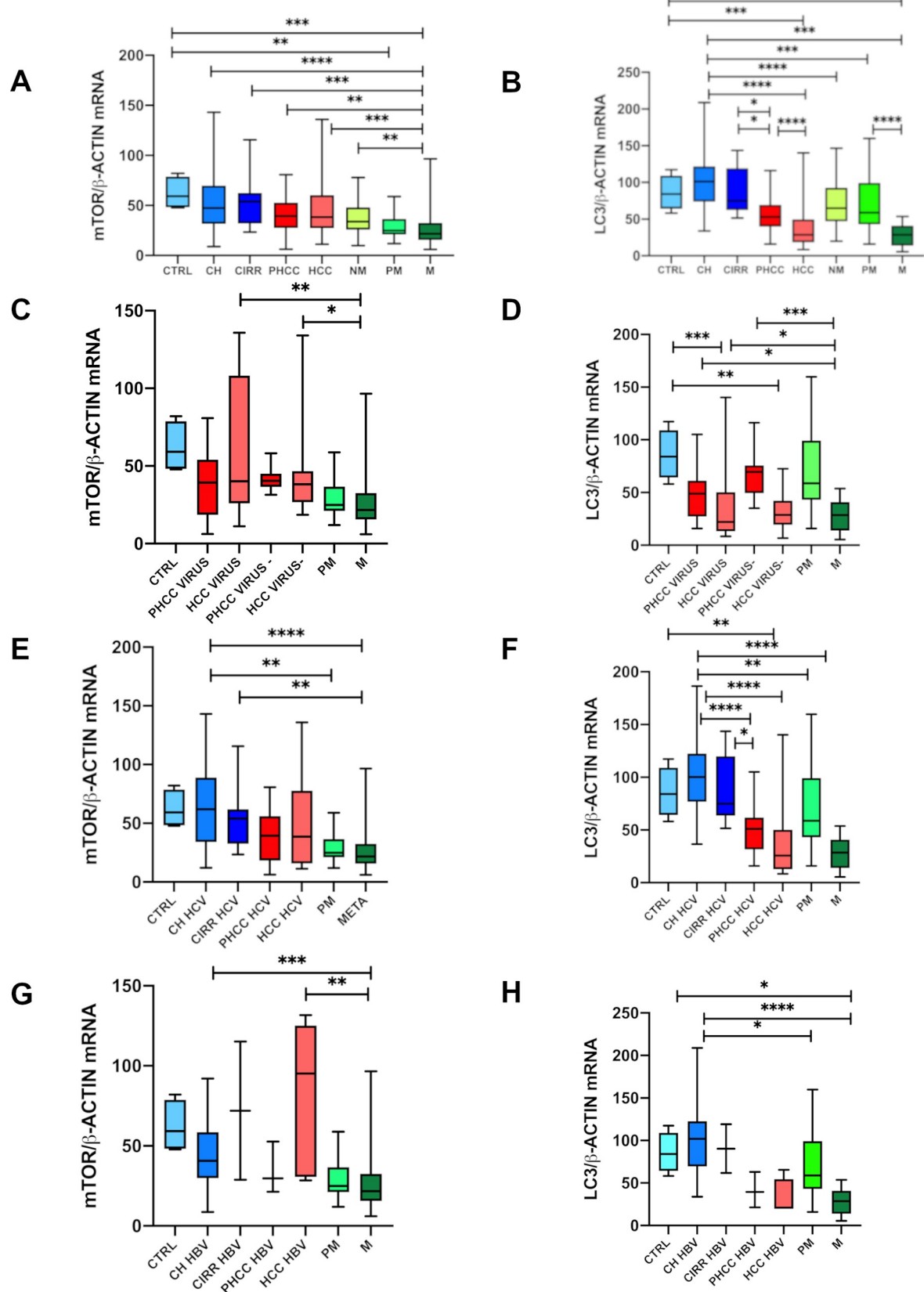

**Fig 1.** Gene expression analysis of mTOR/β-actin (A,C,E,G) and LC3/β-actin (B,D,F,H) ratios by quantitative absolute Real-Time PCR using SYBR Green. The unknown mTOR, LC3 and β-actin mRNA amounts in the samples were extrapolated by the respective standard curves performed using serial dilution (1:10) from $10^8$ to $10^2$ copies/μl of a reference sample. From each sample, the normalized amount of the mTOR and LC3 was obtained by the ratio of mTOR and LC3 to β-actin, used as housekeeping gene. Data are represented as box plot: Box is drawn between the first and third quartiles and the line marks the median, the bar the min to max value. The results of mTOR/β-actin ratios and LC3/β-actin ratios were reported considering: **A, B**) all liver tissues in relation to the pathology vs controls; **C, D**) tissues separated according to the presence (Virus) or absence (Virus-) of viral infection; **E, F**) tissues with HCV-related disease; **G, H**) tissues with HBV-related disease. *CTRL*: Control group, tissues obtained from cholecystectomy and surrounding hyperplasia. *CH*: Chronic hepatitis; *CIRR*: Cirrhosis; *PHCC*: Cirrhotic tissues surrounding hepatocellular carcinoma; *HCC*: Hepatocellular carcinoma; *HCV*: Tissue from hepatitis C virus-related; *HBV*: Tissue from hepatitis B virus-related; *Virus*: All tissues with viral infection; *Virus-*: All tissues without viral infection; *NM*: Histologically normal liver tissues resected far as possible from the metastatic nodule; *PM*: Liver tissues surrounding the metastatic nodule; *M*: Metastatic nodules. $P = 0.05$:*; $P = 0.005$: **; $P = 0.0001$:***; $P < 0.00001$: ****.

**LC3 Gene expression.** A statistically significant difference ($P < 0.0001$, Kruskal Wallis test) emerged in this parameter in the groups considered.

LC3 transcript levels were higher in CTRL (86.1±23) than in HCC (36.7±27; q = 0.003) and M (27.9±15; q = 0.0008).

The highest LC3 gene expression observed in CH (100.2±37) was statistically significant compared with PHCC (55.2±25; q = 0.0007), HCC (36.7±27; q = 0.0007) and all secondary tumorous tissues: NM (69.7±31; q = 0.004), PM (68.6±36; q = 0.003) and M (27.9±15; q = 0.0007). Therefore, statistically significant differences were found between CIRR (87.42 ±30) and PHCC (q = 0.03), HCC (q = 0.0007), and M (q = 0.0007).

All tissues surrounding both primary and secondary tumors showed a significant higher expression of LC3 mRNA levels compared to the tumorous tissues (PHCC vs HCC: q = 0.03 and PM vs M: q = 0.0007). No difference in LC3 transcripts was found between HCC and M. (Fig 1B).

According to viral infection, significantly lower LC3 mRNA levels were observed in both virus infected HCC (Virus HCC: 35.3±32) and no-virus infected HCC (Virus- HCC: 39.4±16)

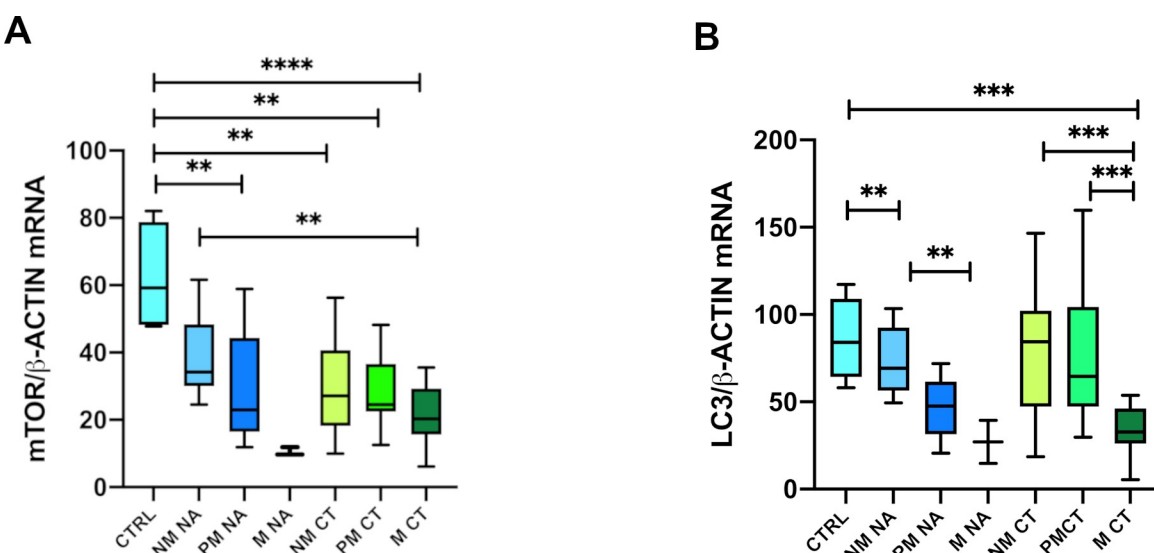

**Fig 2. mTOR/β-actin and LC3/β-actin controls ratios in metastatic tissues of treated and untreated patients before the surgery. A**) results of mTOR/β-actin gene expression. **B**) results of LC3/β-actin gene expression. *CRTL*: Control group. *NM NA*: Tissues resected far as possible from the metastatic nodule of patients that not received adjuvant chemotherapy (NA) before the surgery; *PM NA*: Liver tissues surrounding the metastatic nodule of patients that not received adjuvant chemotherapy before the surgery. *M NA*: Metastatic nodule of patients that not received adjuvant chemotherapy before the surgery. *NA*: No therapy before the surgery. *CT*: Adjuvant chemotherapy before the surgery. $P = 0.05$:*; $P = 0.005$: **; $P = 0.0001$:***; $P < 0.00001$: ****.

## p-mTOR(Ser2448) and mTOR PROTEIN EXPRESSION

**A**  p-mTOR(Ser2448)  289 kDa
mTOR  289kDa
Beta-actina  45kDa

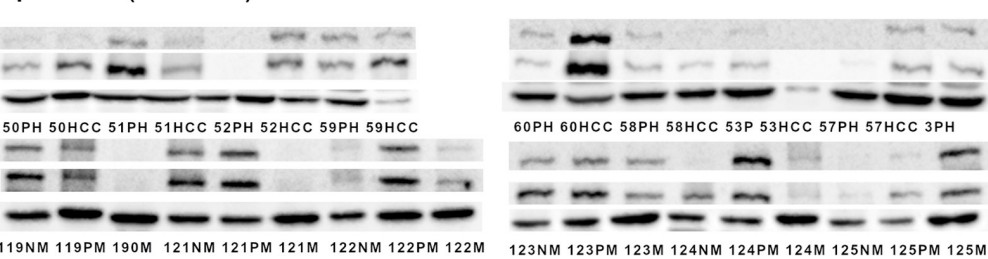

50PH 50HCC 51PH 51HCC 52PH 52HCC 59PH 59HCC

60PH 60HCC 58PH 58HCC 53P 53HCC 57PH 57HCC 3PH

**B**  p-mTOR(Ser2448) 289 kDa
mTOR  289 kDa
Beta-actin  45 kDa

119NM 119PM 190M 121NM 121PM 121M 122NM 122PM 122M

123NM 123PM 123M 124NM 124PM 124M 125NM 125PM 125M

## p-ULK1(Ser757) and ULK1 PROTEIN EXPRESSION

**C**  p-ULK1(Ser792)  140 kDa
ULK1 150 kDa
Beta-actin  45 kDa

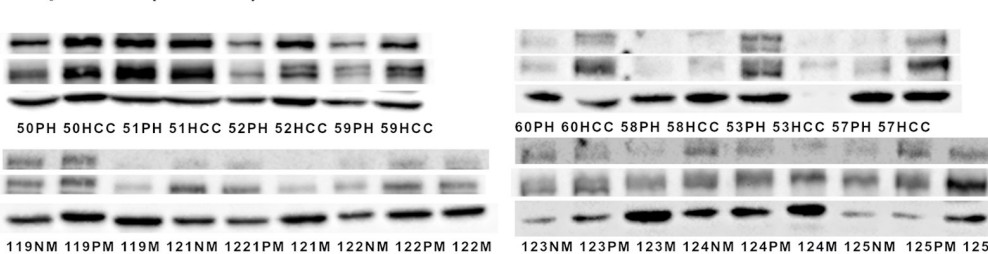

50PH 50HCC 51PH 51HCC 52PH 52HCC 59PH 59HCC

60PH 60HCC 58PH 58HCC 53PH 53HCC 57PH 57HCC

**D**  p-ULK1(Ser792)  140 kDa
ULK1  150 kDa
Beta-actin 45 kDa

119NM 119PM 119M 121NM 1221PM 121M 122NM 122PM 122M

123NM 123PM 123M 124NM 124PM 124M 125NM 125PM 125M

## p-RAPTOR(Ser792) and RAPTOR PROTEIN EXPRESSION

**E**  p-RAPTOR(Ser792) 150 kDa
RAPTOR 150
Beta-actin  45 kDa

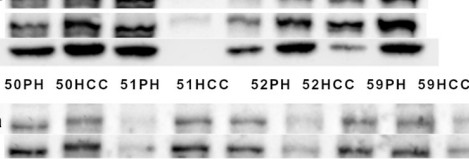

50PH 50HCC 51PH  51HCC  52PH 52HCC 59PH 59HCC

60PH 60HCC 58PH 58HCC 53PH 53HCC 57PH 57HCC

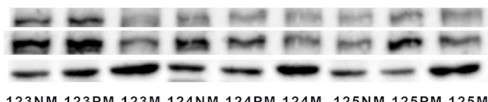

**F**  p-RAPTOR(Ser792) 150 kDa
RAPTOR 150 kDa
Beta-actin 45 kDa

119NM 119PM 119M 121NM 121PM 121M 122NM 122PM 122M

123NM 123PM 123M 124NM 124PM 124M 125NM 125PM 125M

## LC3-II  and LC3-I PROTEIN EXPRESSION

**G**  LC3-II 16 kDa
LC3-I 14 kDa
Beta-actin 45kDa

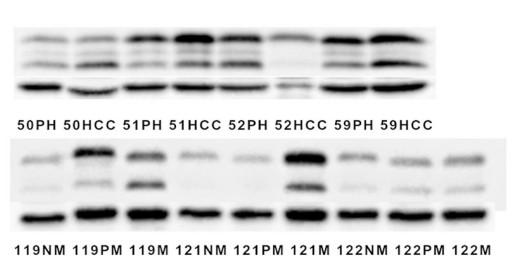

50PH 50HCC 51PH 51HCC 52PH 52HCC 59PH 59HCC

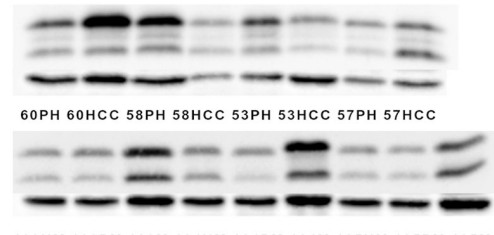

60PH 60HCC 58PH 58HCC 53PH 53HCC 57PH 57HCC

**H**  LC3-II 16kDa
LC3-I 14 kDa
Beta-actin  45kDa

119NM 119PM 119M 121NM 121PM 121M 122NM 122PM 122M

123NM 123PM 123M 124NM 124PM 124M 125NM 125PM 125M

## p62 PROTEIN EXPRESSION

**I**  p62  62kDa
Beta-actin 45kDa

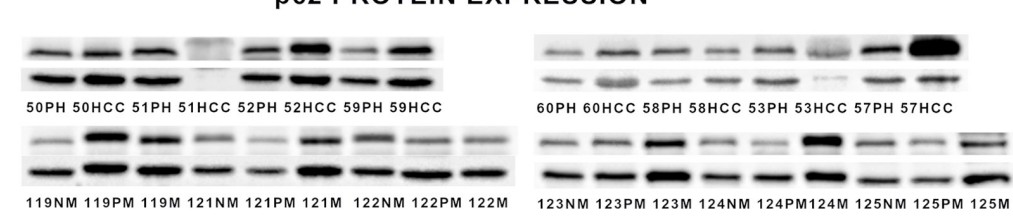

50PH 50HCC 51PH 51HCC 52PH 52HCC 59PH 59HCC

60PH 60HCC 58PH 58HCC 53PH 53HCC 57PH 57HCC

**L**  p-62 62kDa
Beta-actin 45 kDa

119NM 119PM 119M 121NM 121PM 121M  122NM 122PM 122M

123NM 123PM 123M 124NM 124PM124M 125NM 125PM 125M

**Fig 3.** Western Blot analysis representative bands relative to: p-mTOR(Ser2448), mTOR and β-actin protein expression of patients with primary liver cancer (A) and with liver metastases from CRC (B); p-ULK(Ser757), ULK1 and β-actin protein expression of patients with primary liver cancer (C) and liver metastases from colorectal cancer (D); p-Raptor(Ser792), Raptor and β-actin protein expression of patients with primary liver cancer (E) and liver metastases from colorectal cancer (F); LC3-I, LC3-II and β-actin protein expression of patients with primary liver cancer (G) and liver metastases from colorectal cancer (H); p62 and β-actin protein expression of patients with primary liver cancer (I) and liver metastases from colorectal cancer (L). For p-mTOR(Ser2448), mTOR, p-ULK (Ser757), ULK1, Raptor analysis, given their molecular weight of 289, 140 and 150 KDa respectively, the protein lysate was loaded onto 7.5%

SDS-PAGE, onto 10% for p62 KDa and for LC3-I, LC3-II and β-actin analysis, the protein lysate was loaded onto 15% SDS-PAGE, given their molecular weights of 16KDa, 14KDa and 45KDa respectively. PH: Liver tissues surrounding hepatocellular carcinoma; HCC: Hepatocellular carcinoma; *NM*: Histologically normal liver tissues resected far as possible from the metastatic nodule; *PM*: Liver tissues surrounding the metastatic nodule; *M*: Metastatic nodules.

when compared with CTRL (q = 0.003 and q = 0.03 respectively). Statistically significant differences were also observed between Virus PHCC (48.8±23) and Virus- PHCC (72.1±23) compared to M (q = 0.03 and q = 0.002 respectively) (Fig 1D).

In HCV-related tissues, no differences emerged between CRTL, CH and CIRR, but a significant difference was found between CTRL and HCC (q = 0.006). A statistically significant reduction in LC3 transcripts levels was seen going from CH (102.1±33.8) to PHCC (50.5 ± 24.8; q = 0.001), HCC (36.5±35.8; q = 0.0007), PM (q = 0.008) and M (q = 0.0007) (Fig 1F).

In HBV-infected tissues, LC3 mRNA levels were significantly higher in CTRL than in HCC (31.5±22; q = 0.03). HBV-related CH (98.39±41) tissues had higher LC3 transcript levels than HCC (31.5±22; q = 0.006), PM (q = 0.03) and M tissues (q = 0.0007) (Fig 1H).

CTRL tissues showed significantly higher levels compared to both MNA and M CT (27.02 ±17; q = 0.02 and 34.1±13.7; q = 0.005 respectively).

In metastatic nodules, no statistically significant differences were found in relation to treatment. In untreated tissues a decrease of LC3 mRNAs emerged between NM NA (74.4±19) vs PM NA (46.7± 18) and between NM NA vs M NA (27.02±12; q = 0.04). While in treated tissues statistically significant differences were found between both NM CT (78.2±38.4) and PM CT(78.5±40) vs M CT (q = 0.004 and q = 0.004) (Fig 2B).

## Protein expression

The protein expression was performed in liver tissue obtained from surgical samples. The small size of biopsies did not allow both mRNA and protein analysis. Western blot was done using proteins extracted from n.191 liver tissues: n. 36 PHCC; n. 30 HCC; n. 40 NM; n. 45 PM; n. 40 M.

**P-mTOR (Ser 4228)/mTOR protein expression.** The Western Blot images of p-mTOR (Ser2448), mTOR and the corresponding ß-actin are shown in Fig 3A and 3B.

The p-mTOR (Ser2448) and mTOR ratio (phosphorylated and non- phosphorylated forms), pointing to the activity of the protein, was considered for statistical analysis.

A statistically significant difference (P<0.0009 by Kruskal-Wallis test) was observed among the patients groups included in the study.

M showed the highest p-mTOR (Ser2448)/mTOR ratio (0.507 ± 0.2) that was statistically significant compared to PHCC (0.40±0.5; q = 0.03) and HCC (0.39±0.5; q = 0.03) (Fig 4A).

Virus-related PHCC (0.42±0.6) and HCC (0.41±0.5) showed statistically significant differences of p-mTOR/mTOR protein expression when compared to M (0.50±0.2; q = 0.03 and q = 0.03 respectively) (Fig 4B).

According to HCV- and HBV-infection status, a statistically significant difference (q = 0.03) only emerged between HBV-related HCC (0.26±0.3) tissues and M (Fig 4C).

Chemotherapy induced a decrease of p-mTOR/mTOR ratio in PM CT (0.308± 0.2) compared to PM NA (0.645±0.3; q = 0.03) and in M CT (0.43± 0.05) vs M NA (0.76± 0.29) without the latter reaching a statistical significance (Fig 4D).

**P-ULK1 (Ser757)/ULK1 protein expression.** Western blot analysis was done in 18 PHCC, 15 HCC, 23 NM, 28P M and 24M. The Western Blot images of p-ULK (Ser757), ULK1 and the corresponding ß-actin are shown in Fig 3C and 3D.

A statistically significant difference (P<0.0001, Kruskal Wallis test) emerged among the groups considered.

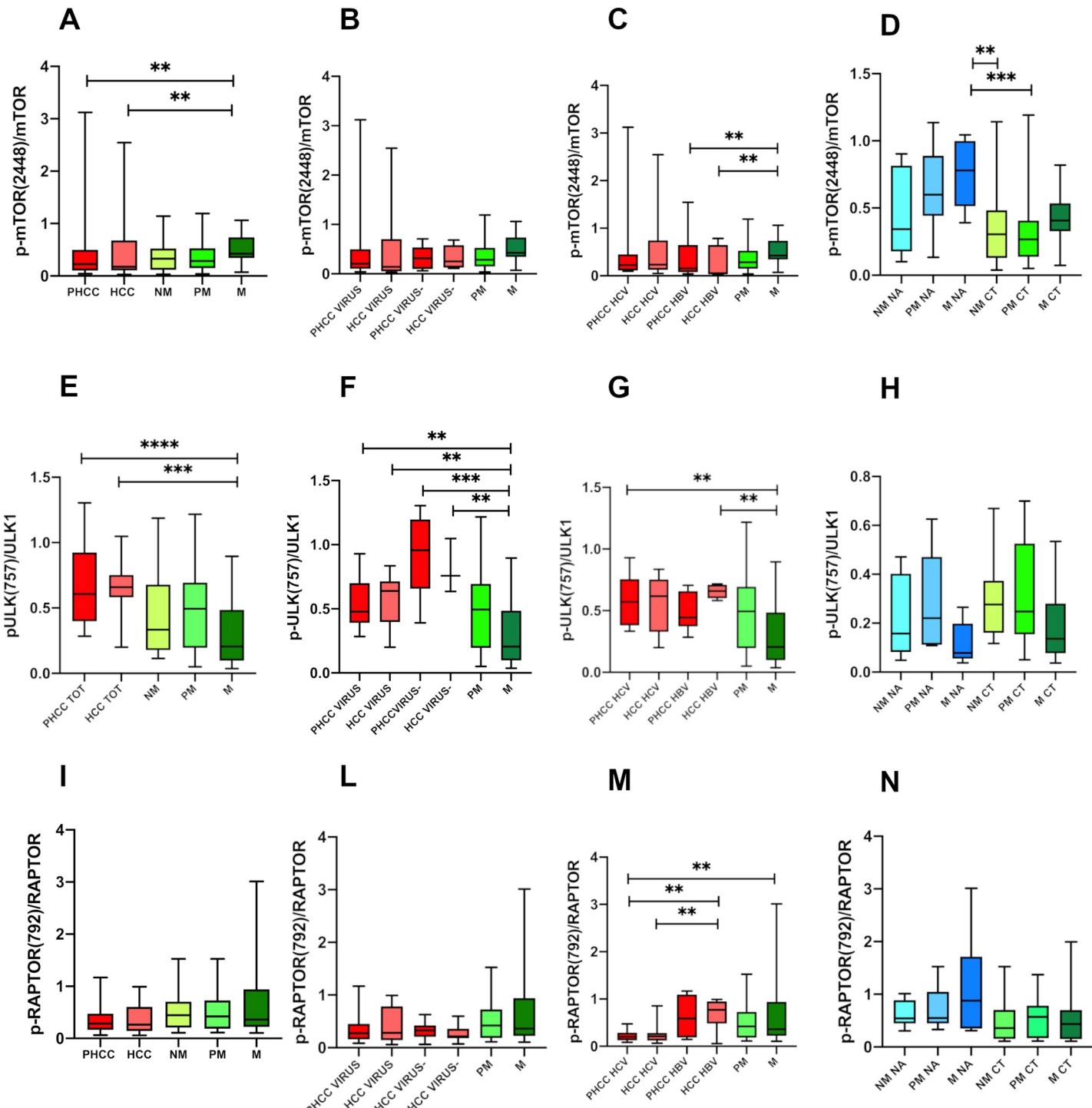

**Fig 4.** Quantitative Western Blot analysis of p-mTOR/mTOR (A,B,C,D), p-ULK(Ser4757)(E,F,G,H) and p-Raptor(Ser792) (I,L,M,N) ratio. Bands intensity were quantified by optical density using Uvitec Alliance Instrument and Uvitec Alliance software. Values are reported as box plot. The results were reported considering: **A, E, I**) primary and secondary tumours and in tissues surrounding tumors; **B, F, L**) tissues separated according to the presence (Virus) or absence (Virus-) of viral infection; **C, G, M**) tissues with HCV-related and HBV-related disease; **D, H, N**) metastatic tissues of treated (CT) and untreated (NA) patients before the surgery. *PHCC*: Cirrhotic tissues surrounding hepatocellular carcinoma; *PHCC HCV*: Cirrhotic tissues surrounding hepatocellular carcinoma HCV-related. *PHCC HBV*: Cirrhotic tissues surrounding hepatocellular carcinoma HBV-related. *HCC*: Hepatocellular carcinoma; *HCC HCV*: Hepatocellular carcinoma HCV-related. *HCC HBV*: Hepatocellular carcinoma HBV-related. *NM*: Histologically normal liver tissues resected far as possible from the metastatic nodule; *PM*: Liver tissues surrounding the metastatic nodule; *M*: Metastatic nodules.

PHCC (0.65±0.2) and HCC (0.63±0.2) showed higher p-ULK1(Ser757)/ULK1 protein levels compared to M (0.30±0.2; q = 0.02 and q = 0.003 respectively) (Fig 4E).

The low p-ULK1(Ser757)/ULK1 expression observed in M was statistically significant compared to both virus and no virus-related PHCC and HCC [Virus PHCC (0.54±0.1), q = 0.02; Virus-PHCC (0.93±0.03), q = 0.003; Virus HCC (0.58±0.1); q = 0.01); Virus- HCC (0.81±0.2), q = 0.02; respectively] (Fig 4F).

Moreover, statistically significant differences between M and both HCV- PHCC related (0.59± 0.2) and HBV- HCC (0.65±0.05) related infection were found (q = 0.03 and q = 0.02, respectively) (Fig 4G).

No significant differences were observed in metastatic tissues related to chemotherapy (Fig 4H).

**P-Raptor (Ser792)/Raptor protein expression.** The phosphorylation of raptor is required for mTORC1 kinase activity. The Western Blot images of p-ULK (Ser757), ULK1 and the corresponding ß-actin are shown in Fig 3E and 3F.

Therefore for statistical analysis the phosphor-Raptor(Ser792)/Raptor ratio was considered.

The Kruskal-Wallis test revealed a statistically significant difference (P<0.0001) among the various groups.

No statistically significant differences were observed among PHCC, HCC, NM, PM, M (Fig 4I) neither in relation to the presence nor in absence of viral infection (Fig 4L).

In HCV- related HCC, phospho-Raptor(Ser792)/Raptor protein expression was significantly lower with respect to HBV-related HCC (0.28±0.2 vs 0.68±0.3; q = 0.04).

A statistically significant difference also emerged between HCV-PHCC (0.229±0.1) and both HBV-related PHCC (0.64±0.4; q = 0.04) and HCC (q = 0.04) and M (0.66±0.6; q = 0.04) (Fig 4M).

No significant differences were observed in metastatic tissues related to chemotherapy (Fig 4N).

**LC3-II/LC3-I protein expression.** The Western Blot images of LC3-I, LC3-II and the corresponding β-actin are shown in Fig 3G and 3H.

For statistical analysis, the LC3II/lC3I ratio, which reflects the successful autophagosome formation, was considered.

A statistically significant difference (P<0.0001 by Kruskal-Wallis test) was observed between the patient groups included in the study.

Metastatic nodules expressed the highest (0.66±0.3) protein levels compared with HCC (0.35±0.4; q = 0.0006) and PHCC (0.21±0.1; q = 0.0006). A statistically significant difference in LC3-II/LC3-I protein levels was found in PM (0.49±0.3) compared with PHCC (q = 0.0006) (Fig 5A).

In both Virus and Virus- related diseases, PM tissues showed higher LC3II/lC3I ratio levels compared with both Virus PHCC (0.22±0.2; q = 0.0006) and Virus- PHCC (0.15±0.1; q = 0.004). Statistically significant differences also emerged between M tissues compared with Virus HCC (0.39±0.4; q = 0.004) or Virus- HCC (0.16±0.2; q = 0.002) and when PHCC tissues were compared with M (Virus PHCC vs M: q = 0.0006; Virus- PHCC vs M: q = 0.001) (Fig 5B).

According to C virus infection: statistically significant differences in LC3II/lC3I levels were observed in HCV-PHCC (0.17±0.1) compared to PM (q = 0.0006) and M (q = 0.0006).

However HCV-related HCC showed lower LC3II/lC3I levels (0.34±0.3) than the PM (q = 0.04) and M (q = 0.009). No difference in LC3II/lC3I levels was seen between tissues with HBV-related infection and metastatic tissues (Fig 5C).

Analyzing metastatic tissues, both NM NA (0.290±0.2) and PM NA (0.261±0.1) showed lower LC3-II/LC3-I protein levels than M NA (0.474±0.2; q = 0.03 in both), PM CT (0.2611 ±0.3; q = 0.05 and q = 0.04 respectively) and M CT (0.70±0.6; q = 0.02 in both) (Fig 5D).

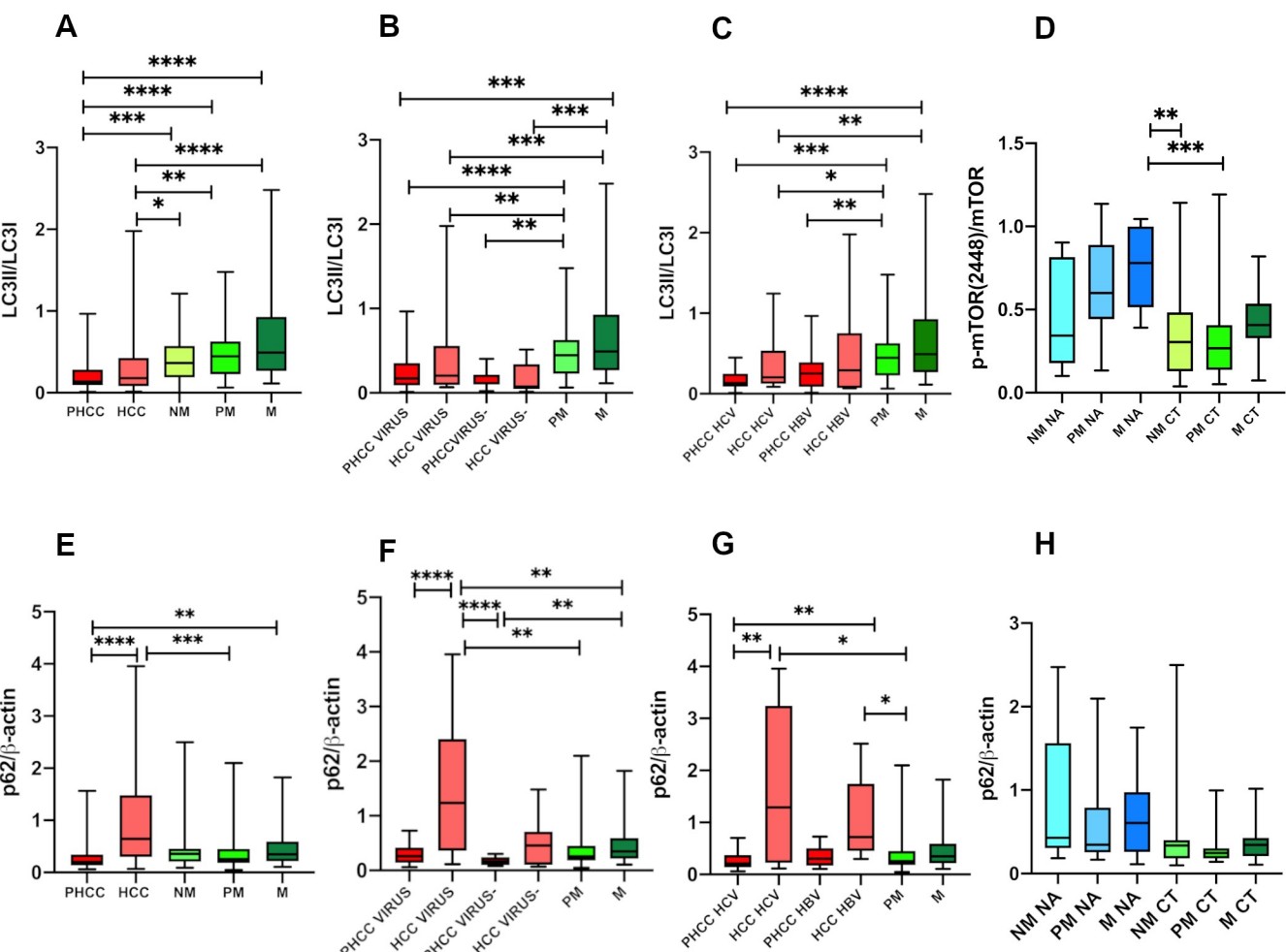

**Fig 5.** Quantitative Western Blot analysis of LC3-II/LC-3I, (A,B,C,D) and p62 (E,F,G,H) ratio. Bands intensity were quantified by optical density using Uvitec Alliance Instrument and Uvitec Alliance software. Values are reported as box plot. The results were reported considering: **A, E**) primary and secondary tumours and in tissues surrounding tumours; **B, F**) tissues separated according to the presence (Virus) or absence (Virus-) of viral infection; **C, G**) tissues with HCV-related and HBV-related disease; **D, H**) metastatic tissues of treated (CT) and untreated (NA) patients before the surgery. *PHCC*: Cirrhotic tissues surrounding hepatocellular carcinoma; *PHCC HCV*: Cirrhotic tissues surrounding hepatocellular carcinoma HCV-related. *PHCC HBV*: Cirrhotic tissues surrounding hepatocellular carcinoma HBV-related. *HCC*: Hepatocellular carcinoma; *HCC HCV*: Hepatocellular carcinoma HCV-related. *HCC HBV*: Hepatocellular carcinoma HBV-related. *NM*: Histologically normal liver tissues resected far as possible from the metastatic nodule; *PM*: Liver tissues surrounding the metastatic nodule; *M*: Metastatic nodules.

**p62/β-actin protein expression.** The Western blot images of p62 protein, considered a marker of autophagy flux, are show in Fig 3I and 3L.

Significant differences in p62 proteins levels were found among the patient groups considered (P<0.0001 by Kruskal Wallis test).

HCC tissues expressed the significantly highest p62 levels (1.16±1.1) compared to PHCC (0,263±0.1; q = 0.0003) and PM (0.37±0.3; q = 0.006), while PHCC showed the lowest p62 levels statistically significant respect to M (0.47±0.4; q = 0.03). No differences were observed between NM (0.5±0.6), PM (0.37±0.3) and M (Fig 5E).

According to the presence of viral infection, in Virus HCC (1.44±1.2) the p62 levels were statistically significantly higher compared to both Virus and Virus- PHCC (0.30±0.2 and 0.17±0.07; q = 0.001 and q = <0.0002 respectively), and to all metastatic tissues: NM (q = 0.02),

PM (q = 0.001), M (q = 0.02). The lowest p62 levels found in Virus- PHCC were statistically significant compared to NM (q = 0.006) and M (q = 0.01) (Fig 5F).

Statistically significant differences were observed between HCV-related PHCC and both HCV- and HBV- related HCC (0.27±0.2 vs 1.60±1.4 and vs 1.11±0.8; q = 0.006 in both) and between both HCV- and HBV-related HCC versus PM (q = 0.01 in both) (Fig 5G).

No significant differences were found in metastatic tissues related to chemotherapy (Fig 5H).

## Correlations

Correlations between LC3 and mTOR transcripts and proteins calculated by Spearman correlation were reported in Table 2.

Significant positive correlations emerged between LC3 and mTOR mRNA transcripts in all groups of tissues considered. These correlations were also significant when tissues were separated according to HCV- and HBV- infection in CH tissues and related to HCV in CIRR, PHCC, HCC tissues.

Regarding proteins, a significant inverse correlation between LC3 and mTOR emerged in HCC (p = 0.034) and in HBV-related HCC tissues (P = 0.0003). Inverse correlations were found between p-ULK(Ser757) and p-Raptor(Ser72) proteins in overall tissues surrounding tumors (P = 0.013) and in tumors (P = 0.023). These inverse correlations were significant in all metastatic tissues (NM: P = 0.012; PM; P = 0.0005; M: P = 0.018) and related to chemotherapy in tissues surrounding metastatic nodules (NM CT: P = 0.037 and PM CT: P = 0.040).

## Discussion

Most liver diseases are linked to a context of viral infection and tissue injury in which different cell types, cell interactions and cell activation degrees coexist, then the evaluation of autophagy status in these tissues is then required.

### mTOR expression

Deregulated mTOR signaling has therefore been associated with human diseases as well as with the development of cancer [29].

In our study, no differences in mTOR transcripts levels were found from chronic hepatitis to cirrhosis and HCC with respect to mRNA observed in controls and NM. These data denote that mTOR mRNA expression is unrelated to the progression of liver disease from CH to CIRR and HCC, and that it is independent of C and B virus infection. In fact, in our series, all patients with chronic hepatitis and cirrhosis were HCV- and HBV-related.

It is known that during infection, viruses seek nutrition to survive and often exploit host mechanisms that control cellular metabolic processes [30]. Moreover, viruses can subvert the host metabolism by targeting mTOR complexes to gain a replicative advantage, while host cells may regulate the mTOR pathway to facilitate virus clearance. Based on timing, cell type and pathogens, alterations in mTOR signaling can therefore have beneficial or harmful consequences for the host.

Different "in vitro models" based on B and C virus infected cell lines demonstrated a dominant role of viral protein in the modulation of the mTOR pathway. mTORC1 is essential for HCV RNA replication and for new particle production [31]. In HCV infection, NS5A, a nonstructural protein of and a crucial factor in viral replication, can activate mTOR through the PI3K/Akt pathway by directly binding to the p85 subunit of PI3K, or impair the combination between mTOR and FKBP38 (an immunosuppressant FK506-binding protein) to block apoptosis [32].

**Table 2. Correlation analysis among mTOR and LC3 gene and protein expression, p62, Raptor and ULK protein expression.**

| | mTOR–LC3 mRNA | | mTOR–LC3 PROTEIN | | mTOR–p62 PROTEIN | | LC3-p62 PROTEIN | | LC3–Raptor PROTEIN | | LC3–ULK1 PROTEIN | | mTOR–Raptor PROTEIN | | mTOR–ULK1 PROTEIN | | p62–Raptor PROTEIN | | p62–ULK1 PROTEIN | | ULK1–Raptor PROTEIN | |
|---|---|---|---|---|---|---|---|---|---|---|---|---|---|---|---|---|---|---|---|---|---|---|
| | Corr | P-val | Corr | P-val | Corr | P-val | Corr | P-val | Corr | P-val | Corr | P-val | Corr | P-val | Corr | P-val | Corr | P-val | Corr | P-val | Corr | P-val |
| Overall NT | 0.56 | **<0.00001** | -0.31 | 0.057 | 0.07 | 0.567 | 0.14 | 0.241 | 0.02 | 0.893 | 0.11 | 0.552 | -0.19 | 0.161 | 0.07 | 0.710 | -0.10 | 0.452 | 0.21 | 0.248 | -0.37 | **0.013** |
| Overall PT | 0.45 | **0.0005** | -0.11 | 0.382 | -0.08 | 0.550 | -0.15 | 0.260 | -0.12 | 0.419 | -0.02 | 0.915 | 0.14 | 0.354 | -0.08 | 0.656 | -0.00 | 0.998 | -0.03 | 0.883 | -0.36 | **0.023** |
| Overall T | 0.32 | **0.022** | -0.08 | 0.547 | 0.03 | 0.897 | 0.44 | **0.030** | 0.10 | 0.640 | 0.46 | 0.154 | -0.33 | 0.151 | 0.35 | 0.331 | -0.13 | 0.600 | 0.52 | 0.133 | 0.03 | 0.914 |
| PHCC | 0.62 | **0.0008** | 0.09 | 0.673 | 0.36 | 0.106 | -0.05 | 0.831 | -0.02 | 0.943 | 0.07 | 0.880 | -0.47 | 0.072 | -0.10 | 0.810 | -0.02 | 0.954 | -0.42 | 0.232 | 0.12 | 0.710 |
| HCC | 0.48 | **0.009** | -0.44 | **0.034** | 0.24 | 0.165 | -0.16 | 0.344 | 0.05 | 0.786 | -0.05 | 0.825 | -0.08 | 0.671 | 0.18 | 0.428 | -0.12 | 0.478 | 0.08 | 0.728 | -0.53 | **0.012** |
| NM | 0.44 | **0.017** | -0.31 | 0.057 | 0.12 | 0.446 | -0.13 | 0.396 | -0.19 | 0.239 | 0.12 | 0.592 | -0.15 | 0.363 | 0.26 | 0.245 | -0.34 | 0.054 | 0.16 | 0.463 | -0.65 | **0.0005** |
| PM | 0.38 | **0.036** | -0.24 | 0.119 | 0.15 | 0.402 | 0.06 | 0.725 | -0.18 | 0.327 | 0.13 | 0.565 | 0.29 | 0.109 | 0.29 | 0.198 | 0.02 | 0.918 | -0.18 | 0.430 | -0.51 | **0.018** |
| M | 0.09 | 0.679 | -0.02 | 0.900 | 0.66 | **0.044** | -0.50 | 0.143 | 0.09 | 0.811 | -0.66 | 0.175 | -0.25 | 0.492 | 0.83 | 0.058 | -0.42 | 0.232 | 0.71 | 0.136 | -0.09 | 0.919 |
| NM NA | -0.43 | 0.299 | -0.64 | 0.054 | 0.69 | **0.035** | -0.50 | 0.143 | -0.15 | 0.682 | 0.26 | 0.658 | -0.55 | 0.104 | 0.60 | 0.242 | -0.29 | 0.427 | 0.60 | 0.242 | -0.77 | 0.103 |
| PM NA | | | -0.38 | 0.279 | 0.05 | 0.892 | -0.14 | 0.707 | 0.05 | 0.892 | -0.09 | 0.919 | 0.25 | 0.492 | 0.26 | 0.658 | 0.02 | 0.973 | 0.54 | 0.297 | -0.49 | 0.356 |
| M NA | | | 0.41 | 0.247 | 0.19 | 0.411 | -0.09 | 0.719 | 0.02 | 0.940 | 0.09 | 0.797 | -0.08 | 0.743 | 0.45 | 0.173 | -0.44 | 0.056 | 0.22 | 0.521 | -0.65 | **0.037** |
| NM CT | 0.84 | **<0.00001** | -0.25 | 0.290 | -0.11 | 0.641 | 0.11 | 0.649 | -0.08 | 0.761 | -0.14 | 0.694 | -0.26 | 0.317 | 0.26 | 0.435 | -0.22 | 0.384 | 0.04 | 0.924 | -0.64 | **0.040** |
| PM CT | 0.66 | **0.012** | 0.11 | 0.657 | -0.14 | 0.575 | -0.05 | 0.843 | -0.24 | 0.361 | -0.20 | 0.584 | 0.34 | 0.192 | 0.24 | 0.485 | -0.23 | 0.372 | -0.31 | 0.356 | -0.13 | 0.744 |
| M CT | 0.43 | 0.146 | -0.11 | 0.662 | | | | | | | | | | | | | | | | | | |
| CH | 0.51 | **<0.0001** | | | | | | | | | | | | | | | | | | | | |
| CH HCV+ | 0.54 | **<0.00001** | | | | | | | | | | | | | | | | | | | | |
| CH HBV+ | 0.53 | **<0.0002** | | | | | | | | | | | | | | | | | | | | |
| CIRR | 0.78 | **0.004** | | | | | | | | | | | | | | | | | | | | |
| PHCC HCV+ | 0.88 | **0.0001** | | | 0.19 | 0.564 | 0.46 | 0.104 | 0.02 | 0.977 | 0.80 | 0.133 | -0.31 | 0.462 | 0.20 | 0.783 | 0.25 | 0.595 | 0.60 | 0.350 | 0.83 | 0.058 |
| HCC HCV+ | 0.62 | **0.026** | | | 0.43 | 0.166 | 0.08 | 0.817 | 0.31 | 0.564 | 0.40 | 0.750 | -0.31 | 0.564 | -0.40 | 0.750 | -0.40 | 0.750 | | | 0.10 | 0.950 |
| PHCC HBV+ | | | -0.05 | 0.935 | 0.43 | 0.354 | 0.54 | 0.236 | -0.48 | 0.121 | 0.70 | 0.233 | -0.16 | 0.634 | 0.20 | 0.917 | -0.60 | 0.073 | -0.31 | 0.564 | 0.18 | 0.713 |
| HCC HBV+ | | | -1.00 | **0.0003** | 0.10 | 0.950 | -0.10 | 0.950 | -0.09 | 0.811 | | | -0.58 | 0.088 | | | -0.06 | 0.892 | | | 0.20 | 0.714 |

Corr: Spearman's correlation coefficient; P-val: P-value.

Moreover, HCV seems to be particularly involved in the activation of autophagy, considering that it interacts with lipid metabolism [33] to affect virion assembly and maturation, although HCV induces autophagosome accumulation, but does not improve protein degradation in liver biopsies [34].

Also in HBV infection, the HBx protein, which transactivates viral and cellular genes by interacting with nuclear transcription factors, is able to activate PI3K/Akt-mTOR to promote persistent, non-cytopathic HBV replication [35], while pre-S1 can activate the Akt/mTOR pathway through up regulation of VEGFR-2 [36].

In our series, the lack of statistically significant differences between both no virus- and virus-related PHCC and HCC and between both HCV- and HBV-related PHCC and HCC seems to indicate an independent role of virus infection in the mTOR mRNA expression. During viral infection and liver damage different cell types, cell interaction and degree of cell activation coexist therefore, in this context, other molecular pathways may be involved in the regulation of mTOR.

Significant differences in mTOR transcript levels were found between primary and secondary liver tissues, in both tissues surrounding tumors and tumors, with the lowest mTOR gene expression observed in M. These downregulations observed in metastatic tissues indicate a dual role of mTOR pathway in the modulation of cell proliferation in liver tumors. Contrary to what is reported by other Authors [37], we did not find differences between PHCC and HCC tissues.

In primary tumors mTOR can confer many growth advantages to cancer cells or progenitor stem cells [38], such as promoting cell proliferation and resistance to apoptosis. In addition, mTOR can regulate telomerase activity in hepatocarcinogenesis or may indirectly induce tumorigenesis by the suppression of autophagy, which plays a crucial role in tumor suppression by eliminating damaged cells. Moreover, the lowest mTOR gene expression observed in M indicates a phase-specific function of mTOR. It is important to remember the biological differences between metastatic cells arising from the clonal expansion of primary colon-rectal cancer cells [39] versus transformed hepatocytes of primary tumors. Colorectal liver metastases appear to be highly subjected to mutations in the Akt/mTOR pathway [40], resulting in deregulation of mTOR.

Furthermore, the microenvironment is a determining factor in the modulation of gene expression and cell signaling in metastases versus primary tumors. The energy deficit, genotoxic stress and oxygen deprivation present in HCC undoubtedly operate on the activation of TSC1 and TSC2, with a consequent inhibition of mTOR. Our data might confirm the important role of the micro environment; in fact, no statistically significant difference in mTOR gene expression was found between HCC arising in normal liver and M, while statistically significant differences were found between HCC arising in cirrhotic HCV- and HBV-related tissues and metastatic liver cells from colon-rectal cancer that colonized a "normal liver". In this regard, in gene expression studies "normal liver tissues" to use as controls are not only difficult to obtain (for ethical reasons) but are also difficult to define. Generally, unaffected hepatic tissues have been used as representative of "normal liver tissues" simply based on their normal morphology such as tissues surrounding tumors, both primary and metastatic. The higher mTOR gene expression seen in CTRL tissues by comparison with "normal" tissues surrounding tumors (such as non-cirrhotic tissues surrounding primary cancer, NM NA and PM NA), probably reflects an involvement of cancer cells, particularly evident in the tissues neighboring to the tumor mass, in the downregulation of this molecular pathway. Regarding mTOR protein expression, in this study we observed an increase of S2448 phosphorylation of mTOR proteins (active form of mTOR associated with mTORC1) in M with respect to HCC, but no differences between HCC and tissues surrounding HCC were found, contrary to that reported

by other authors [41]. The increased levels of the P-mTOR protein observed in M may be related to the promotion of the anabolic processes involved in cell proliferation and survival. Indeed, mTORC1 could negatively regulate autophagy through its direct phosphorylation and suppression of the kinase complex ULK1/Atg13/FIP200 that is required for the initiation of autophagy [42]. Moreover, variation in mTOR protein expression was found in the tissues of patients treated and not treated with systemic therapy before CRC resection. Our data indicate an upregulation of the mTOR pathway in PM and M of patients not treated with chemotherapy, and a downregulation in those treated. Cytotoxic drugs have been shown to increase Raptor levels [43]; in our series, moreover, the low p-mTOR/mTOR protein levels found in tissues of CRC patients treated might point to an inhibitory role of the mTOR signal pathway in these tissues. At present, mTOR inhibitors and their combination with other drugs are available for the treatment of the whole spectrum of malignancies [44–48]. Our samples do not include HCC tissue from patients treated with mTOR inhibitor, something that warrants further investigation.

## ULK1 expression

ULK1 complex plays an essential role in the initiation stage of autophagy. ULK1 can be phosphorylated by mTORC1 or AMPK to prevent or promote autophagy [49]. Under nutrient-rich conditions, the mTORC1 active form phosphorylates ULK1 (Ser757) disrupting ULK1-AMPK interaction, thus making ULK1 inactive [23,50].

In this study, phosphor-ULK1(Ser757)/ULK1 levels, which represent the inactive form of ULK1, were higher in HCC and surrounding HCC tissues with respect to metastatic nodule, confirming the activation-initiation of autophagy in secondary liver tumors. While no differences in autophagy activation were observed between virus and no virus-infected tissues, autophagy was less activated only in HBV-related HCC compared to M. An immunohistochemistry study demonstrated that the ULK1 overexpression in HCC and adjacent peritumoral tissues, without a statistically significant difference between the two groups, was related to tumor size and worse survival time [51].

Although we analyzed the inhibition of ULK1 no differences were observed between PHCC and HCC tissues indicating a comparable degree of autophagy activation in these tissues.

## Raptor expression

Raptor is a scaffold protein recruited for the identification and binding mTORC1 substrates. This protein dissociates from the complex in response to amino acids and other stimuli. Therefore, Raptor acts a sensor to respond to changes in the micro-environment through the phosphorylation of its different sites mediated by distinct kinases that regulate mTORC1 function. Moreover, during nutrient withdrawal and energy stress AMP-activated protein kinase (AMPK), through the binding of ULK phosphorylates Raptor at Ser792 site inhibiting mTORC1 activity [18].

In our series we found an inverse correlation between p-Raptor (Ser792) and p-ULK (Ser757) in all metastatic tissues especially in tissues surrounding metastatic nodules of treated patients. Therefore, these data point to a relevant role of Raptor phosphorylation at Ser792 in liver metastases during chemotherapy. Nevertheless, significant differences in phospho-Raptor (Ser792)/Raptor protein levels were observed between tissues related to C and B virus infection. Compared to HCV-related HCC, HBV-related HCC showed an increase of both phospho-Raptor (Ser792)/Raptor protein levels and ULK1(Ser757)/ULK1 expression, suggesting a different activation of autophagy related to the diverse C and B virus proteins.

Furthermore, it has been proven that Raptor overexpression enhances replication of HCV. Currently, no data on viral load are available, in our tissues to support that.

## LC3 expression

In order to evaluate autophagy, we analyzed the gene and protein expression of LC3, which represents a suitable marker of autophagic activity [52,53]. We found a downregulation of LC3 transcripts related to the severity of liver disease, from CH to PHCC and HCC, regardless of HCV or HBV-infection, and low levels in primary and secondary tumours. It is interesting to observe the low LC3 transcript levels in both HCV- and HBV-related HCC and M. These data could be a consequence of an involvement of post-transcriptional or post-translational regulators, such as microRNAs, small non-coding RNAs, that can be overexpressed or downregulated in different phases of liver damage. Many evidences indicate that miRNAs, by acting either as oncogenes or tumour suppressors, are involved in the process of HCC tumorigenesis and development through the modulation of the autophagic signalling networks [54,55]. miR-181a was reported to repress autophagy in HCC by targeting pro-autophagic protein Atg5, leading to the reduction of apoptosis of HCC cells and acceleration of hepatoma growth [56]. A significant decrease in miR-7, which induces autophagy by targeting mTOR, was observed in tumour tissue compared to normal tissue suggesting its potential antitumor role in HCC [57]. He et al. demonstrated that miR-21 inhibits autophagy via the Akt/mTOR signalling pathway [58].

No differences in LC3 transcripts were observed between CTRL and all normal tissues surrounding both primary and secondary tumours.

Moreover, the progressive decrease in LC3 transcripts levels from CRTL to NM NA and PM NA, (statistically significant between CTRL vs PM NA) indicated a progressive autophagosome formation in these histologically normal tissues related to the adjacency of tumour. Therefore, these molecular changes observed in tissues without apparent morphological abnormality point to a serious selection of the control group that should be carried out considering only those tissues, when available, resected far from the surrounding metastatic nodule tissues and untreated.

LC3 protein expression is considered the only protein marker reliably associated with completed autophagosomes. The high LC3II/LC3I protein levels found in all metastatic tissues compared to primary liver cancer tissues denote a different degree of autophagosome formation in these tissues.

Moreover, in a previous study [59] a low expression of Beclin-1, one of the main autophagic factor that bridges autophagy, apoptosis and differentiation, in HCC tissues was found.

Therefore, our data attest that autophagy is mainly activated in liver metastases and downregulated in primary liver cancer.

Regarding viral infection, it is interesting to observe that only in HCV-related tissues the LC3II/LC3I protein levels were significantly lower compared to metastatic tissues, indicating a greater involvement of C virus in the formation of autophagosomes with respect to B virus infection.

In fact, it is known that HCV replication is compartmentalized by lipid bilayer membranes and in infected hepatocytes, HCV replication may induce accumulation of autophagosome, which would interfere with autophagy itself [33]. Recently, the high LC3 expression, by immunohistochemistry analysis, both in the tumour and non-tumour liver was significantly associated with lower HCC recurrence in patients who underwent curative hepatectomy for HCC [60]. Therefore, LC3 was also considered a potential marker for predicting HCC recurrence and overall survival [61,62].

The higher LC3II/LC3I expression found in all metastatic tissues with respect to with primary liver cancer tissues confirms the findings of Yuan-Fei Peng et al. [32], although the metastases they considered derive from HCC and not from CRC.

The lack of significant differences in the LC3II/LC3I protein levels observed in metastatic nodules of patients treated and untreated and the higher level in PM CT compared to PM NA indicate a prominent role of chemotherapy in the activation of autophagy in non-tumorous tissues adjacent to the metastatic nodule.

Moreover, the positive correlation between mTOR and LC3 transcripts found in CH, CIRR, PHCC, HCC, NM and PM point to a strong interconnection of these markers, particularly evident in infected tissues. In spite of mRNA levels, the correlation analysis of protein expression showed inverse correlations only in HBV-related HCC and in histologically normal tissue far from the metastatic nodule (NM), indicating the different role of autophagy in both primary and secondary tumors, and also in metastases of patients treated and untreated with chemotherapy before surgery. In HCC, autophagy mainly seems to be a tumor suppressing process, while in metastases it seems to be useful for tumor cell survival, as seen by the effect of chemotherapeutic agents. The contribution of mTOR inhibitors administered with conventional chemotherapy drugs may therefore be different in the treatment of primary, HBV- and HCV-related infection, and of secondary liver tumors. In fact, it is known that HCC cells are prone to developing multidrug resistance, due to the heterogeneity of the tumor and the fragility of its genome [63].

**p62/SQSTM1 expression.** p62 is a selective autophagy receptor that interacts with both several ubiquitinated substrates and multiple sites on LC3 within autophagosome, where it is sequestered and degraded. Since p62 acts as a substrate during autophagy, degradation is considered an indicator of autophagic flux. Its levels usually inversely correlate with autophagic degradation [64].

In this study, we found the highest p62 amount in HCC virus-related tissues when compared to M, while we found statistically significant lower p62 levels in PHCC both HCV- and HBV-related when compared to HCC. Therefore, the p62 protein accumulations indicate an inhibition of the autophagy flux in primary tumor, but not in cirrhotic tissues surrounding HCC, PM and M.

p62 controls many cellular processes with or without the involvement of autophagy [27]. For example, ROS can induce p62 expression trough the Kaep-NRF2 pathway, licensing the induction of p62-mediated selective autophagy. In tumor tissues, damaged organelles and misfolded proteins are aberrantly expressed; therefore, this metabolic stress leads to p62 accumulation, that is critical for tumorigenesis process, process more relevant in primary liver tumor in respect to metastasis.

In vitro experiments demonstrated that HBV X protein induces autophagy but, in the same time, in order to enhance viral DNA replication, reduces protein degradation in the autolysosome [34]. Viral non-structural proteins and viral RNA polymerase of HCV, can interact with the host proteins required for autophagy and then modulate their replication.

As reported above, in HBV-related HCC, Raptor and ULK1 are more expressed, indicating the intricate interconnection between these proteins in the modulation of autophagy strictly linked to tissue microenvironment.

## Normal liver tissues

The drawback of the liver study is the difficulty to obtain, for ethical reasons, a "genuinely" normal human liver. In this study, we analyzed the gene expression of mTOR and LC3 in so-called histologically normal liver tissues, such as tissues obtained during cholecystectomy and surrounding hyperplasia, tissues surrounding HCC arising in non-cirrhotic liver and tissues surrounding secondary liver tumors at the different distances from the metastatic nodule.

We expected to find a similar amount of transcript in all the tissues considered as "normal".

Surprisingly, our data showed a decrease of both mTOR and LC3 mRNAs in all normal and no- treated tissues surrounding tumors when compared to CTRL, even if a statistically significant difference was found only between CTRL and PM NA. Nevertheless, we think that PM tissues cannot be representative of normal liver tissues.

## Conclusion

In our study, the activated form of mTOR protein is upregulated in metastases and downregulated in HCC, suggesting its prominent role as an anabolic protein, which promotes tumor growth and proliferative processes. On the other hand, the low LC3-II/LC3-I protein levels also found in HCC tissues might be linked to the proliferation and growth of tumor masses, favoring invasion processes.

Our data show that in metastases, the increased p-mTOR (Ser2448) levels, marker of mTOR activation, may be responsible for autophagy repression inhibiting ULK1 by phosphorylating ULK1 on Ser757.

The reduced levels of ULK1-phosphorylation at Ser757 found in these tissues seems to point at an increase of autophagy activity as a consequence of Raptor phosphorylation at Ser792 site, responsible for inhibition mTOR activity. These intricate interactions could explain the increased LC3 levels and the low p62 levels that represent the autophagy flux.

Therefore, autophagy in metastatic nodules might represent a protective way for metastatic cell survival in a hostile microenvironment.

In HCC tissues, we observed a reduction of mTOR activation associated whit an increase of expression of ULK1, suggesting an activation of autophagy-initiation. Interestingly, the different expression of Raptor phosphorylation, high in HBV-related and low in HCV-related tissues suggest the involvement of other kinase activation processes also related to the different viral proteins.

Nevertheless, the decreased LC3 and the increased of p62 levels found in both HCV- and HBV- related HCC, with respect to metastases, indicate an impairment of the autophagy flux important for the promotion of tumorigenesis process.

The analysis of mTOR and LC3 in the metastatic tissues of patients who undergo chemotherapy treatment before surgery, compared to the metastatic tissues of patients who do not undergo treatment, points to a decisive role of chemotherapy in the activation of autophagic processes related to the mTOR pathway in all metastatic tissues considered, and to LC3-II/LC3-I only in tissues surrounding metastases. In these tissues, autophagy could be useful for the survival of cancerous cells, reducing necrosis and promoting neoplastic cell growth. In the early phases of liver disease, virus infection is not crucial in the transcription of TOR and LC3.

Last, but not least, in gene expression studies the selection of control group is a determining factor for the results comparison. The conflicting results reported in the literature on the gene expression studies can be explained not only by the use of different methods, but also by having compared data on "pathological" tissues with different so-called "normal" controls.

## Supporting information

**S1 Fig.**
(DOCX)

**S1 Table. Power analysis of the study.**
(XLSX)

**S2 Table. Raw data.**
(PDF)

## Acknowledgments

We would like to thank Romilda Cardin for the collection of biopsies from patients with chronic liver diseases; Lara Borsetto for the collection of surgical tissues from patients with HCC; Maura Digito for the storage of samples from patients with CRC in the Biobanks of the First Surgical Clinic–DISCOG; Chiara Carlotto and Milena Minotto for real time PCR analysis.

## Author Contributions

**Conceptualization:** Marina Bortolami.

**Data curation:** Marina Bortolami, Clara Benna, Isacco Maretto.

**Formal analysis:** Clara Benna, Andrea Errico.

**Funding acquisition:** Marina Bortolami, Fabio Farinati.

**Investigation:** Alessandra Comparato.

**Methodology:** Alessandra Comparato, Andrea Errico.

**Resources:** Isacco Maretto, Salvatore Pucciarelli, Umberto Cillo.

**Supervision:** Fabio Farinati.

**Validation:** Andrea Errico.

**Writing – original draft:** Marina Bortolami.

**Writing – review & editing:** Clara Benna.

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
