## [Decision Letter · Decision Letter 0]

19 Aug 2020

PONE-D-20-07836

Gene and protein expression of mTOR and LC3 in hepatocellular carcinoma, colorectal liver metastasis  and "normal" liver tissues

PLOS ONE

Dear Dr. Bortolami,

Thank you for submitting your manuscript to PLOS ONE. After careful consideration, we feel that it has merit but does not fully meet PLOS ONE’s publication criteria as it currently stands. Therefore, we invite you to submit a revised version of the manuscript that addresses the points raised during the review process.

Academic Editor: To make valid conclusions about the autophagy status, the authors should assess the levels of selective autophagy targets such as SQSTM1/p62 or NBR1, as well as the activation (phosphorylation) of key autophagy signalling molecules such as ULK1. Also, as the p values have already been shown in the figures, there is no need to repeat them in the text, it makes the manuscript difficult to follow.

Reviewer 2

This is a well performed study showing the levels of mTOR and LC3 in hepatocellular carcinoma, colorectal liver metastasis and "normal" liver tissues. Previous works show the important role that autophagy can have in liver diseases. In this study, the authors show the different levels of protein and mRNA expression in mTOR and LC3 that primary and secondary hepatic tumors present. I have no substantial objections to the experimental work, which is straightforward and presented in a clear manner. As such, these data support the conclusions. However, for completeness, there are a number of points that should be addressed by the authors.

Minor points:

•     In Table 1 (Etiological characteristics of subject), in the HCC column it indicates that the number of patients is 57 but when separated by gender they add up to 56 (20 female and 36 male)

•     In line 268 of the document, the paragraph talks about Figure 1D but indicates Figure 1C

•     In the document I would add some information that helps your understanding:

-     In line 282 of the document the CIRR value is not indicated

-     In line 305 of the document the value of M is not indicated

-     At the end of the paragraph that begins to describe figure 2B I would add Fig 2B, on line 301. 

•     In Figure 1A, we can see the difference in the errors of CH and CIRR, but in the document on line 254 it indicates that CH (53.3±30), CIRR (56.2±30). check

•     In Figure 4 A and B, we can see that the PHCC and HCC values are close to 0.2, but in line 320 of the document it indicates values of  (0.4±0.5) or (0.39±0.5) respectively. Check

•     In Table 2 there are two questions that I would like to clarify:

-     First in the mRNA correlation column of mTOR and LC3 in CH HBV + there is no coefficient value but it does have p value.

-     Second, I do not understand the meaning of negative values greater than 1, such as the NM NA value of the same column as before. Spearman's correlation coefficient values are between 1 and -1.

•     As the authors explain that the M samples present the lowest levels of mRNA of all the samples in the mTOR and LC3 gene, but then they observe the highest expression values of both proteins in M.

•     Figure 3 presents 9 representative samples of all the samples studied that are included in the densitometers of figure 4. I believe that the films Western blots of all tumors can be displayed on platforms such as (Mendeley Data) for this to be consulted. (Justification Point 3)

•     The discussion talks about autophagy (line 519 of the document), but the document does not provide enough data to determine the state of autophagy, in my opinion it is difficult to determine the state of autophagy based on two genes or proteins only. It is more convenient to indicate the possible activation of autophagy, or something similar.

•     Finally, the quality of the images of all the figures is very low, which makes it difficult to understand the text. I think that they should be replaced by higher quality images that allow greater visibility.

We look forward to receiving your revised manuscript.

Kind regards,

Vladimir Trajkovic

Academic Editor

PLOS ONE

Journal Requirements:

2.We noticed minor instances of text overlap with the following previous publication(s), which need to be addressed:

(1) https://clincancerres.aacrjournals.org/content/21/22/5037.full

The text that needs to be addressed involves the Introduction section (lines 93-96)

In your revision please ensure you cite all your sources (including your own works), and quote or rephrase any duplicated text outside the methods section. Further consideration is dependent on these concerns being addressed.

3. Please provide additional details regarding participant consent. In the ethics statement in the Methods and online submission information, please ensure that you have specified what type of consent you obtained (for instance, written or verbal, and if verbal, how it was documented and witnessed).

4. In your Methods section, please provide additional information about the participant recruitment method and the demographic details of your participants. Please ensure you have provided sufficient details to replicate the analyses such as: a) the recruitment date range (month and year), b) a description of any inclusion/exclusion criteria that were applied to participant recruitment, c) a description of how participants were recruited.

5. Please provide a sample size and power calculation in the Methods, or discuss the reasons for not performing one before study initiation.

6.Thank you for including your ethics statement: 'The Ethical University Hospital Ethics Committee approved the study protocol (no.47081, CESC code 3312/AO/14).'    

(a) Please amend your current ethics statement to include the full name of the ethics committee/institutional review board(s) that approved your specific study.  

(b) Once you have amended this/these statement(s) in the Methods section of the manuscript, please add the same text to the “Ethics Statement” field of the submission form (via “Edit Submission”).

7. PLOS requires an ORCID iD for the corresponding author in Editorial Manager on papers submitted after December 6th, 2016. Please ensure that you have an ORCID iD and that it is validated in Editorial Manager. To do this, go to ‘Update my Information’ (in the upper left-hand corner of the main menu), and click on the Fetch/Validate link next to the ORCID field. This will take you to the ORCID site and allow you to create a new iD or authenticate a pre-existing iD in Editorial Manager. Please see the following video for instructions on linking an ORCID iD to your Editorial Manager account: https://www.youtube.com/watch?v=_xcclfuvtxQ

Reviewers' comments:

Reviewer's Responses to Questions

**Comments to the Author**

1. Is the manuscript technically sound, and do the data support the conclusions?

Reviewer #1: No

Reviewer #2: Yes

2. Has the statistical analysis been performed appropriately and rigorously? 

Reviewer #1: No

Reviewer #2: Yes

3. Have the authors made all data underlying the findings in their manuscript fully available?

Reviewer #1: No

Reviewer #2: No

4. Is the manuscript presented in an intelligible fashion and written in standard English?

Reviewer #1: No

Reviewer #2: Yes

5. Review Comments to the Author

Reviewer #1: The manuscript of Bortolami et al. contains a series of data in the form of statistical comparisons which are difficult to follow. No experimental data are reported at least for some representative tests.

Reviewer #2: This is a well performed study showing the levels of mTOR and LC3 in hepatocellular carcinoma, colorectal liver metastasis and "normal" liver tissues. Previous works show the important role that autophagy can have in liver diseases. In this study, the authors show the different levels of protein and mRNA expression in mTOR and LC3 that primary and secondary hepatic tumors present. I have no substantial objections to the experimental work, which is straightforward and presented in a clear manner. As such, these data support the conclusions. However, for completeness, there are a number of points that should be addressed by the authors.

Minor points:

• In Table 1 (Etiological characteristics of subject), in the HCC column it indicates that the number of patients is 57 but when separated by gender they add up to 56 (20 female and 36 male)

• In line 268 of the document, the paragraph talks about Figure 1D but indicates Figure 1C

• In the document I would add some information that helps your understanding:

- In line 282 of the document the CIRR value is not indicated

- In line 305 of the document the value of M is not indicated

- At the end of the paragraph that begins to describe figure 2B I would add Fig 2B, on line 301.

• In Figure 1A, we can see the difference in the errors of CH and CIRR, but in the document on line 254 it indicates that CH (53.3±30), CIRR (56.2±30). check

• In Figure 4 A and B, we can see that the PHCC and HCC values are close to 0.2, but in line 320 of the document it indicates values of (0.4±0.5) or (0.39±0.5) respectively. Check

• In Table 2 there are two questions that I would like to clarify:

- First in the mRNA correlation column of mTOR and LC3 in CH HBV + there is no coefficient value but it does have p value.

- Second, I do not understand the meaning of negative values greater than 1, such as the NM NA value of the same column as before. Spearman's correlation coefficient values are between 1 and -1.

• As the authors explain that the M samples present the lowest levels of mRNA of all the samples in the mTOR and LC3 gene, but then they observe the highest expression values of both proteins in M.

• Figure 3 presents 9 representative samples of all the samples studied that are included in the densitometers of figure 4. I believe that the films Western blots of all tumors can be displayed on platforms such as (Mendeley Data) for this to be consulted. (Justification Point 3)

• The discussion talks about autophagy (line 519 of the document), but the document does not provide enough data to determine the state of autophagy, in my opinion it is difficult to determine the state of autophagy based on two genes or proteins only. It is more convenient to indicate the possible activation of autophagy, or something similar.

• Finally, the quality of the images of all the figures is very low, which makes it difficult to understand the text. I think that they should be replaced by higher quality images that allow greater visibility.

6. PLOS authors have the option to publish the peer review history of their article (what does this mean?). If published, this will include your full peer review and any attached files.

Reviewer #1: No

Reviewer #2: No

---

## [Author Response · Author response to Decision Letter 0]

29 Oct 2020

Response to Reviewers

Dear Editor,

we thank the Academic Editor and the Reviewers for the careful attention and the comments given to our manuscript PONE-D-20-07836 “Gene and protein expression of mTOR and LC3 in hepatocellular carcinoma, colorectal liver metastasis and "normal" liver tissues”. We thoroughly followed their suggestions and we feel that the manuscript ultimately improved. We hope that the present revised version of our manuscript fully meets the criteria of Plos One and that will be considered for publication.

Here a point by point answer to Academic Editor and the Reviewer #2

Academic Editor: To make valid conclusions about the autophagy status, the authors should assess the levels of selective autophagy targets such as SQSTM1/p62 or NBR1, as well as the activation (phosphorylation) of key autophagy signaling molecules such as ULK1.

Answer: We thank the Academic Editor for the comments and for giving us the opportunity to resubmit a revised version of the manuscript. We agree with the suggestion, therefore, we assessed the protein expression of p62, ULK1 and Raptor in the studied tissues. We added the corresponding paragraphs in the results session and in the discussion section. All the inserted new paragraphs are written in blue.

Also, as the p values have already been shown in the figures, there is no need to repeat them in the text, it makes the manuscript difficult to follow.

Answer: done as requested. We showed the q-values in the text and deleted the P-values.

Reviewer 2

This is a well performed study showing the levels of mTOR and LC3 in hepatocellular carcinoma, colorectal liver metastasis and "normal" liver tissues. Previous works show the important role that autophagy can have in liver diseases. In this study, the authors show the different levels of protein and mRNA expression in mTOR and LC3 that primary and secondary hepatic tumors present. I have no substantial objections to the experimental work, which is straightforward and presented in a clear manner. As such, these data support the conclusions. 

We thank Reviewer 2 for the positive comments.

However, for completeness, there are a number of points that should be addressed by the authors.

Minor points:

• In Table 1 (Etiological characteristics of subject), in the HCC column it indicates that the number of patients is 57 but when separated by gender they add up to 56 (20 female and 36 male)

Answer: done as requested. We corrected the total number.

• In line 268 of the document, the paragraph talks about Figure 1D but indicates Figure 1C

Answer: done as requested. 

• In the document I would add some information that helps your understanding:

- In line 282 of the document the CIRR value is not indicated

Answer: done as requested. We apologize for the oversight.

- In line 305 of the document the value of M is not indicated

Answer: done as requested. We apologize for the oversight.

- At the end of the paragraph that begins to describe figure 2B I would add Fig 2B, on line 301. 

Answer: done as requested. 

• In Figure 1A, we can see the difference in the errors of CH and CIRR, but in the document on line 254 it indicates that CH (53.3±30), CIRR (56.2±30). check

Answer: done as requested. Figure 1 was redone.

• In Figure 4 A and B, we can see that the PHCC and HCC values are close to 0.2, but in line 320 of the document it indicates values of (0.4±0.5) or (0.39±0.5) respectively. Check

Answer: done as requested. Figure 4 was redone.

• In Table 2 there are two questions that I would like to clarify:

- First in the mRNA correlation column of mTOR and LC3 in CH HBV + there is no coefficient value but it does have p value.

Answer: done as requested. We apologize for the oversight.

- Second, I do not understand the meaning of negative values greater than 1, such as the NM NA value of the same column as before. Spearman's correlation coefficient values are between 1 and -1.

Answer: done as requested. We apologize for the typos, all the values greater than 1 where missing the “0,” above.

• As the authors explain that the M samples present the lowest levels of mRNA of all the samples in the mTOR and LC3 gene, but then they observe the highest expression values of both proteins in M.

Answer: Exactly, we observed a decrease of mTOR and LC3 in gene expression with the progression of liver damage, but in order to evaluate the activity of the corresponding proteins we considered the difference between the phosphorylate form of the protein and the total quantity of the protein, independently of its total amount. 

• Figure 3 presents 9 representative samples of all the samples studied that are included in the densitometers of figure 4. I believe that the films Western blots of all tumors can be displayed on platforms such as (Mendeley Data) for this to be consulted. (Justification Point 3)

Answer: we will follow this suggestion.

• The discussion talks about autophagy (line 519 of the document), but the document does not provide enough data to determine the state of autophagy, in my opinion it is difficult to determine the state of autophagy based on two genes or proteins only. It is more convenient to indicate the possible activation of autophagy, or something similar.

Answer: As suggested by the Academic Editor we assessed the protein expression of p62, ULK1 and Raptor in the studied tissues. We added the corresponding paragraphs in the results session and in the discussion section. All the inserted new paragraphs are written in blue.

• Finally, the quality of the images of all the figures is very low, which makes it difficult to understand the text. I think that they should be replaced by higher quality images that allow greater visibility.

Answer: done as requested.

---

## [Decision Letter · Decision Letter 1]

9 Dec 2020

Gene and protein expression of mTOR and LC3 in hepatocellular carcinoma, colorectal liver metastasis  and "normal" liver tissues

PONE-D-20-07836R1

Dear Dr. Bortolami,

We’re pleased to inform you that your manuscript has been judged scientifically suitable for publication and will be formally accepted for publication once it meets all outstanding technical requirements.

Kind regards,

Vladimir Trajkovic

Academic Editor

PLOS ONE

Additional Editor Comments (optional):

Reviewers' comments:

Reviewer's Responses to Questions

**Comments to the Author**

1. If the authors have adequately addressed your comments raised in a previous round of review and you feel that this manuscript is now acceptable for publication, you may indicate that here to bypass the “Comments to the Author” section, enter your conflict of interest statement in the “Confidential to Editor” section, and submit your "Accept" recommendation.

Reviewer #2: (No Response)

2. Is the manuscript technically sound, and do the data support the conclusions?

Reviewer #2: Yes

3. Has the statistical analysis been performed appropriately and rigorously? 

Reviewer #2: Yes

4. Have the authors made all data underlying the findings in their manuscript fully available?

Reviewer #2: Yes

5. Is the manuscript presented in an intelligible fashion and written in standard English?

Reviewer #2: Yes

6. Review Comments to the Author

Reviewer #2: (No Response)

7. PLOS authors have the option to publish the peer review history of their article (what does this mean?). If published, this will include your full peer review and any attached files.

Reviewer #2: **Yes: **Alicia Bort Bueno

---

## [Editor Report · Acceptance letter]

11 Dec 2020

PONE-D-20-07836R1 

Gene and protein expression of mTOR and LC3 in hepatocellular carcinoma, colorectal liver metastasis and “normal” liver tissues 

Dear Dr. bortolami:

I'm pleased to inform you that your manuscript has been deemed suitable for publication in PLOS ONE. Congratulations! Your manuscript is now with our production department. 

Kind regards, 

on behalf of

Prof. Vladimir Trajkovic 

Academic Editor

PLOS ONE